# MEMORY-BASED PARAMETER ADAPTATION

**Pablo Sprechmann\*, Siddhant M. Jayakumar\*, Jack W. Rae, Alexander Pritzel**
**Adrià Puigdomènech Badia, Benigno Uria, Oriol Vinyals**
**Demis Hassabis, Razvan Pascanu, Charles Blundell**
DeepMind
London, UK
```
{psprechmann, sidmj, jwrae, apritzel,
adriap, buria, vinyals,
dhcontact, razp, cblundell}@google.com
```

## ABSTRACT

Deep neural networks have excelled on a wide range of problems, from vision to language and game playing. Neural networks very gradually incorporate information into weights as they process data, requiring very low learning rates. If the training distribution shifts, the network is slow to adapt, and when it does adapt, it typically performs badly on the training distribution before the shift. Our method, Memory-based Parameter Adaptation, stores examples in memory and then uses a context-based lookup to directly modify the weights of a neural network. Much higher learning rates can be used for this local adaptation, reneging the need for many iterations over similar data before good predictions can be made. As our method is memory-based, it alleviates several shortcomings of neural networks, such as catastrophic forgetting, fast, stable acquisition of new knowledge, learning with an imbalanced class labels, and fast learning during evaluation. We demonstrate this on a range of supervised tasks: large-scale image classification and language modelling.

## 1 INTRODUCTION

Neural networks have been proven to be powerful function approximators, as shown in a long list of successful applications: image classification (e.g. Krizhevsky et al., 2012), audio processing (e.g. Oord et al., 2016), game playing (e.g. Mnih et al., 2015; Silver et al., 2017), and machine translation (e.g. Wu et al., 2016). Typically these applications apply batch training to large or near-infinite data sets, requiring many iterations to obtain satisfactory performance.

Humans and animals are able to incorporate new knowledge quickly from single examples, continually throughout much of their lifetime. In contrast, neural network-based models rely on the data distribution being stationary and the training procedure using low learning rates and many passes through the training data to obtain good generalisation. This limits their application to life-long learning or dynamic environments and tasks.

Problems in continual learning with neural networks commonly manifest as the phenomenon of catastrophic forgetting (McCloskey & Cohen, 1989; French, 1999): a neural network performs badly on old tasks having been trained to perform well on a new task. Several recent approaches have proven promising at overcoming this, such as elastic weight consolidation (Kirkpatrick et al., 2017). Recent work in language modelling has demonstrated how popular neural language models may appropriately be adapted to take advantage of rare, recently seen words, as in the neural cache (Grave et al., 2016), pointer sentinel networks (Merity et al., 2016) and learning to remember rare events (Kaiser et al., 2017). Our work generalises these approaches and we present experimental results where we apply our model to both continual or incremental learning tasks, as well as language modelling.

We propose Memory-based Parameter Adaptation (MbPA), a method for augmenting neural networks with an episodic memory to allow for rapid acquisition of new knowledge while preserving

---

*Denotes equal contribution.

Figure 1: Architecture for the MbPA model. Left: Training usage. The parametric network is used directly and experiences are stored in the memory. Right: Testing setting. The embedding is used to query the episodic memory, the retrieved context is used to adapt the parameters of the output network.

the high performance and good generalisation of standard deep models. It combines desirable properties of many existing few-shot, continual learning and language models. We draw inspiration from the theory of complementary learning systems (CLS: McClelland et al., 1995; Leibo et al., 2015; Kumaran et al., 2016), where effective continual, life-long learning necessitates two complementary systems: one that allows for the gradual acquisition of structured knowledge, and another that allows rapid learning of the specifics of individual experiences. As such, MbPA consists of two components: a parametric component (a standard neural network) and a non-parametric component (a neural network augmented with a memory containing previous problem instances). The parametric component learns slowly but generalises well, whereas the non-parametric component rapidly adapts the weights of the parametric component. The non-parametric, instance-based adaptation of the weights is local, in the sense the modification is directly dictated by the inputs to the parametric component. The local adaptation is discarded after the model produces its output, avoiding long term consequences of strong local adaptation (such as overfitting), allowing the weights of the parametric model to learn slowly leading to strong performance and generalisation.

The contributions of our work are: *(i)* proposing an architecture for enhancing powerful parametric models with a fast adaptation mechanism to efficiently cope with changes in the task at hand; *(ii)* establish connections between our method and attention mechanisms frequently used for querying memories; *(iii)* present a Bayesian interpretation of the method allowing a principled form of regularisation; *(iv)* evaluating the method on a range of different tasks: continual learning, incremental learning and data distribution shifts, obtaining promising results.

## 2 MODEL-BASED PARAMETER ADAPTATION

Our models consist of three components: an embedding network, $f_\gamma$, a memory $M$ and an output network $g_\theta$. The embedding network, $f_\gamma$, and the output network, $g_\theta$, are standard parametric (feed forward or recurrent) neural networks with parameters $\gamma$ and $\theta$, respectively. The memory $M$ is a dynamically-sized memory module that stores key and value pairs, $M = \{(h_i, v_i)\}$. Keys $\{h_i\}$ are given by the embedding network. The values $\{v_i\}$ correspond to the desired output $y_i$. For classification, $y_i$ would simply be the true class label, whereas for regression, $y_i$ would be the true regression target. Hence, upon observing the $j$-th example, we append the pair $(h_j, v_j)$ to the memory $M$, where:

$$h_j \leftarrow f_\gamma(x_j),$$
$$v_j \leftarrow y_j.$$

The memory has a fixed size and acts as a circular buffer: when it is full, the oldest data is overwritten first. Retrieval from the memory $M$ uses $K$-nearest neighbour search on the keys $\{h_i\}$ with Euclidean distance to obtain the $K$ most similar keys and associated values.

Our model is used differently in the training and testing phases. During training, for a given input $x$, we parametrise the conditional likelihood with a deep neural network given by the composition

---

**Algorithm 1** Model-based Parameter Adaptation

---

**procedure** MBPA-TRAIN

    Sample mini-batch of training examples $B = \{(x_b, y_b)\}_b$ from training data.

    Calculate the embedded mini-batch $B' = \{(f_\gamma(x_b), y_b) : x_b, y_b \in B\}$.

    Update $\theta, \gamma$ by maximising the likelihood (1) of $\theta$ and $\gamma$ with respect to mini-batch $B$

    Add the embedded mini-batch examples $B'$ to memory $M$: $M \leftarrow M \cup B'$.

**procedure** MBPA-TEST(test input: $x$, output prediction: $\hat{y}$)

    Calculate embedding $q = f_\gamma(x)$, and $\Delta_{\text{total}} \leftarrow 0$.

    Retrieve $K$-nearest neighbours to $q$ and producing context, $C = \{(h_k^{(x)}, v_k^{(x)}, w_k^{(x)})\}_{k=1}^{K}$.

    **for** each step of MbPA **do**

        Calculate $\Delta_M(x, \theta + \Delta_{\text{total}})$ according to (4)

        $\Delta_{\text{total}} \leftarrow \Delta_{\text{total}} + \Delta_M(x)$.

    Output prediction $\hat{y} = g_{\theta + \Delta_{\text{total}}}(h)$

---

of the embedding and output networks. Namely,

$$p_{\text{train}}(y|x, \gamma, \theta) = g_\theta(f_\gamma(x)). \tag{1}$$

In the case of classification, the last layer of $g_\theta$ is a softmax layer. The parameters $\{\theta, \gamma\}$ are estimated by maximum likelihood estimation. The memory is updated with new entries, as they are seen, however no local adaptation is performed on the model. Figure 1 (left) shows a diagram of the training setting and Algorithm 1 (MbPA-Train) shows the algorithm for updating MbPA during training.

On the other hand, at test time, it temporarily adapts the parameters of the output network based upon the current input and the contents of the memory $M$. That is, it uses the exact same parametrisation as (1), but with a different set of parameters in the output network.

Let the context $C$ of an input $x$ be the keys, values and associated weights of the $K$ nearest neighbours to query $q = f_\gamma(x)$ in the memory $M$: $C = \{(h_k^{(x)}, v_k^{(x)}, w_k^{(x)})\}_{k=1}^{K}$. The coefficients $w_k^{(x)} \propto \text{kern}(h_k^{(x)}, q)$ are weightings of each of the retrieved neighbours according to their closeness to the query $f_\gamma(x)$. $\text{kern}(h, q)$ is a kernel function which, following (Pritzel et al., 2017), we choose as $\text{kern}(h, q) = \frac{1}{\epsilon + \|h - q\|_2^2}$. The parametrisation of the likelihood takes the form,

$$p(y|x, \theta^x) = p(y|x, \theta^x, C) = g_{\theta^x}(f_\gamma(x)), \tag{2}$$

as opposed to the standard parametric approach $g_\theta(f_\gamma(x))$, where $\theta^x = \theta + \Delta_M(x, \theta)$ with $\Delta_M(x, \theta)$ being a contextual (it is based upon the input $x$) update of the parameters of the output network. The MbPA adaptation corresponds to decreasing the weighted average negative log-likelihood over the retrieved neighbours in $C$. Figure 1 (right) shows a diagram of the testing setting and Algorithm 1(MbPA-Test) shows the algorithm for using MbPA during testing.

An interesting property of the model is that the correction $\Delta_M(x, \theta)$ is such that, as the parametric model becomes better at fitting the training data (and consequently the episodic memories), it self-regulates and diminishes. In the CLS theory, this process is referred to as consolidation, when the parametric model can reliably perform predictions without relying on episodic memories.

## 2.1 MAXIMUM A POSTERIORI INTERPRETATION OF MBPA

We can now derive $\Delta_M(x, \theta)$, motivated by considering the posterior distribution on the parameters $\theta^x$. Let $x$ correspond to the input with context $C = \{h_k, v_k, w_k^{(x)}\}_{k=1}^{K}$. The maximum a posteriori over the context $C$, given the parameters obtained after training $\theta$, can be written as:

$$\max_{\theta^x} \log p(\theta^x|\theta) + \sum_{k=1}^{K} w_k^{(x)} \log p(v_k^{(x)}|h_k^{(x)}, \theta^x, x), \tag{3}$$

where the second term is a weighted likelihood of the data in $C$ and $\log p(\theta^x|\theta) \propto -\frac{\|\theta^x - \theta\|_2^2}{2\alpha_M}$ (i.e. a Gaussian prior on $\theta^x$ centred at $\theta$) can be thought as a regularisation term that prevents overfitting. See Appendix D for details of this derivation.

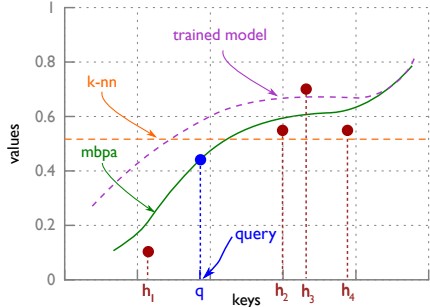

Figure 2: Illustrative diagram of the local fitting on a regression task. Given a query (blue), we retrieve the context from memory showed in red.

Equation (3) does not have a closed form solution, and requires fitting a large number of parameters at inference time. This can be costly and susceptible to overfitting. We can avoid this problem by adapting the reference parameters $\theta$. Specifically, we perform a fixed number of gradient descent steps to minimise (3). One step of gradient descent to the loss in (3) with respect to $\theta^x$ yields

$$\Delta_M(x, \theta) = -\alpha_M \left. \nabla_\theta \sum_{k=1}^{K} w_k^{(x)} \log p(v_k^{(x)} | h_k^{(x)}, \theta^x, x) \right|_\theta - \beta(\theta - \theta^x), \tag{4}$$

where $\beta$ is a scalar hyper-parameter. These adapted parameters are used for output computation but discarded thereafter, as described in Algorithm 1.

## 2.2 FROM ATTENTION TO LOCAL FITTING

A standard formulation of memory augmented networks is in the form of attention (Bahdanau et al., 2014), i.e. query memory to use a weighted average based on some similarity metric.

We can now show that an attention-based procedure is a particular case of local adaptation or MbPA. The details of this are discussed in Appendix E. Effectively, attention can be viewed as fitting a constant function the neighbourhood of memories, whereas MbPA generalises to fit a function parameterised by the output network of our model.

The diagram in Figure 2 illustrates an example in a regression task for simplicity. Given a query (blue), the retrieved memories and their corresponding values are depicted in red. The predictions of an attention based model are shown in orange. We can see that the prediction is biased towards the value of the neighbours with higher functional value. In magenta we represent the predictions made by the model $g_\theta$. We can see that it is not able to explain all memories equally well. This could be either because the problem is too difficult, poor training, or because the a prediction needs to be made while assimilating new information. The green curve show the prediction obtained after adapting the parameters to better explain the episodic memories.

## 3 RELATED WORK

A key component of MbPA is the non-parametric, episodic memory. Many recent works have looked at augmenting neural network systems with memories to allow for fast adaptation or incorporation of new knowledge. Variants of this architecture have been successfully used in the context of classification (Vinyals et al., 2016; Santoro et al., 2016; Kaiser et al., 2017), language modelling (Merity et al., 2016; Grave et al., 2016), reinforcement learning (Blundell et al., 2016; Pritzel et al., 2017), machine translation (Bahdanau et al., 2014), and question answering (Weston et al., 2014), to name a few. For the MbPA experiments below, we use a memory architecture similar to the Differentiable Neural Dictionary (DND) used in Neural Episodic Control (NEC) (Pritzel et al., 2017). One key difference is that we do not train the embedding network through the gradients from the memories (as they are not used at training time).

While many of these approaches share a contextual memory lookup system, MbPA is distinct in the method by which the memories are used. Matching Networks (Vinyals et al., 2016) use a non-

parametric network to map from a few examples to a target class via a kernel weighted average. Prototypical Networks (Snell et al., 2017) extend this and use a linear model instead of a nearest neighbour method.

MbPA is further related to meta-learning approaches for few shot learning. In the context of learning invariant representations for object recognition, Anselmi et al. (2014) proposed a method that can invariantly and discriminatively represent objects using a single sample, even of a new class. In their method, instead of training via gradient descent, image templates are stored in the weights of simple-complex cell networks while objects undergo transformations. Optimisation as a model of few shot learning (Ravi & Larochelle, 2016) proposes using a meta-learner LSTM to control the gradient updates of another network, while Model-Agnostic Meta-Learning (MAML Finn et al. (2017)) proposes a way of doing meta-learning over a distribution of tasks. These methods extend the classic fine-tuning technique used in domain adaptation type of ideas (e.g. fit a given neural network to a small set of new data). The MAML algorithm (particularly related to our work) aims at learning an easily adaptable set of weights, such that given a small amount of training data for a given task following the training distribution, the fine-tuning procedure would effectively adapt the weights to this particular task. Their work does not use any memory or per-example adaptation and is not based on a continual (life-long) learning setting. In contrast, our work, aims at augmenting a powerful neural network with a fine-tuning procedure that is used at inference only. The idea is to enhance the performance of the parametric model while maintaining its full training.

Recent approaches to addressing the continual learning problem have included elastic weight consolidation (Kirkpatrick et al., 2017), where a penalty term is added to the loss for deviations far from previous weights, and learning without forgetting (Li & Hoiem, 2016; Furlanello et al., 2016), where distillation (Hinton et al., 2015) from previously trained models is used to keep old knowledge available. Gradient Episodic Memory for Continual Learning (Lopez-Paz & Ranzato, 2017) attempts to solve the problem by storing data from previous tasks and taking gradient updates when learning new tasks that do not increase the training loss on examples stored in memory.

There has been recent work in applying attention to quickly adapt a subset of fast weights (Ba et al., 2016). A number of recent works in language modelling have augmented prediction with attention over recent examples to account for the distributional shift between training and testing settings. Works in this direction include neural cache (Grave et al., 2016) and pointer sentinel networks (Merity et al., 2016). Learning to remember rare events (Kaiser et al., 2017) augments an LSTM with a key-value memory structure, and meta networks (Munkhdalai & Yu, 2017) combines fast weights with regular weights. Our model shares this flavour of attention and fast weights, while providing a model agnostic memory-based method that applies beyond language modelling.

Works in the context of machine translation relate to MbPA. Gu et al. (2017) explore how to incorporate information from memory into the final model predictions. The authors find that shallow mixing works best. We show in this paper that MbPA is another competitive strategy to shallow mixing, and often working better (PTB for language modelling, ImageNet for image classification). The work by Li et al. (2016) shares the focus on fast-adaptation during inference with our work. Given a test example, the translation model is fine-tuned by fitting similar sentences from the training set. MbPA can be viewed as a generalisation of such approach: it relies on an episodic memory (rather than the training set), contextual lookup and similarity based weighting scheme to fine-tune the original model. Collectively, these allow MbPA to be a powerful domain-agnostic algorithm, which allows it to handle continual and incremental learning.

Finally, we mention that our work is closely related to the local regression and adaptive coefficient models literature, see Loader (2006) and references therein. Locally adaptive methods achieved relatively modest success in high-dimensional classification problems, as fitting many parameters to a few neighbours often leads to over fitting. We attempt to counter this with contextual lookups and a local modification of only a subset of model parameters.

## 4    EXPERIMENTS AND RESULTS

Our scheme unifies elements from traditional approaches to continual, one-shot, and incremental or life-long learning. Models that solve these problems must have certain fundamental attributes

in common: the ability to negate the effects of catastrophic forgetting, unbalanced and scarce data, while displaying rapid acquisition of knowledge and good generalisation.

In essence, these problems require the ability to deal with changes and shifts in data distributions. We demonstrate that MbPA provides a way to address this. More concretely, due to the robustness of the local adaptation, the model can deal with shifts in domain distribution (e.g. train vs test set in language), the task label set (e.g. incremental learning) or sequential distributional shifts (e.g. continual learning). Further, MbPA is agnostic to both task domain (e.g. image or language) and choice of underlying parametric model, e.g. convolutional neural networks (LeCun et al., 1998) or LSTM (Hochreiter & Schmidhuber, 1997).

To this end, our experiments focus on displaying the advantages of MbPA on widely used tasks and datasets, comparing with competing deep learning methods and baselines. We start by looking at the continual learning framework, followed by incremental learning, the problems of unbalanced data and test time distributional changes.

## 4.1 CONTINUAL LEARNING: SEQUENTIAL DISTRIBUTIONAL SHIFT

In this set of experiments, we explored the effects of MbPA on continual learning, i.e. when dealing with the problem of sequentially learning multiple tasks without the ability to revisit a task.

We considered the permuted MNIST setup (Goodfellow et al., 2013). In this setting, each task was given by a different random permutation of the pixels of the MNIST dataset. We explored a chaining of 20 different tasks (20 different permutations) trained sequentially. The model was tested on all tasks it had been trained on thus far.

We trained all models using 10,000 examples per task, comparing to elastic weight consolidation (EWC; Kirkpatrick et al., 2017) and regular gradient descent training. In all cases we rely on a two layer MLP and use Adam (Kingma & Ba, 2014) as the optimiser. The EWC penalty cost was chosen using a grid search, as was the local MbPA learning rate (between 0.0 and 1.0) and number of optimisation steps for MbPA (between 1 and 20).

Figure 3 compares our approach with that of the baselines. For this particular task we worked directly on pixels as our embedding, i.e. $f_\gamma$ is the identity function, and explored regimes where the episodic memory is small. A key takeaway of this experiment is that once a task is catastrophically forgotten, we find that *only a few gradient steps* on carefully selected data from memory are sufficient to recover performance, as MbPA does. Considering the number of updates required to reach the solution from random initialisation, this fact itself might seem surprising. MbPA provides a principled and effective way of performing these updates. The naive approach of performing updates on memories chosen at random from the entire memory is considerably less useful.

We outperformed the MLP, and were superior to EWC for all but one memory size (when storing only a 100 examples per task). Further, the performance of our model grew with the number of examples stored, ceteris paribus. Crucially, our memory requirements are much lower than that of EWC, which requires storing model parameters and Fisher matrices for all tasks seen so far. Unlike EWC we do not store any tasks identifiers, merely appending the memory with a few examples. Further, MbPA does not use knowledge of exact task boundaries or identities of tasks switched to, unlike EWC and other methods. This allows for frequent switches that would otherwise hamper the Fisher calculations needed for models like EWC.

Our method can be combined with any other algorithm such as standard replay from the memory buffer or EWC, providing further improvement. In Figure 3 (right) we combine MbPA and EWC.

## 4.2 INCREMENTAL LEARNING: SHIFTS IN TASK LABEL DISTRIBUTIONS

The goal of this section was to evaluate the model in the context of incremental learning. We considered a classification scenario where a model pre-trained on a subset of classes, was introduced to novel, previously unseen classes. The aim was to incorporate the new related knowledge, as quickly as possible, while preserving knowledge from the previous set. This was as opposed to the continual learning problem where there are distinct tasks without the ability to revisit old data.

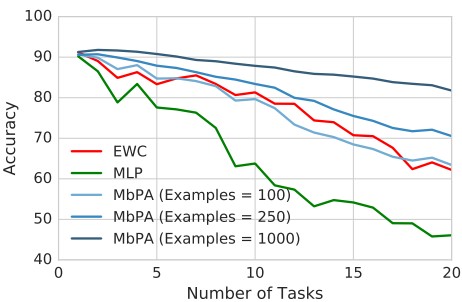 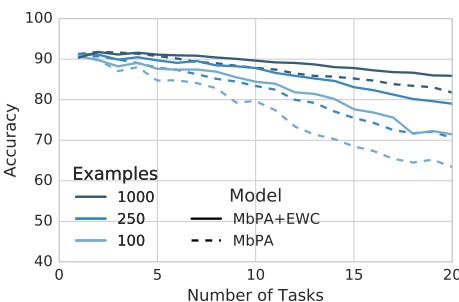

Figure 3: (Left) Results on Permuted MNIST comparing baselines with MbPA using different memory sizes. (Right) Results augmenting MbPA with EWC, showing the flexibility and complementarity of MbPA.

Specifically we considered the problem of image classification on the ImageNet dataset (Russakovsky et al., 2015). As a parametric model we used a ResnetV1 model (He et al., 2016). This was pre-trained on a random subset of the ImageNet dataset containing half of the classes. We then presented all 1000 classes and evaluated how quickly the network can acquire this knowledge (i.e. perform well across all 1000 classes).

For MbPA, we used the penultimate layer of the network as the embedding network $f_\gamma$, forming the key $h$ and query $q$ for our episodic memory $M$. The last fully connected layer was used to initialise the parametric model $g_\theta$. MbPA was applied at test time, using RMSprop with a local learning rate $\alpha_M$ and the number of optimisation steps (as in Algorithm 1) tuned as hyper-parameters.

A natural baseline was to simply fine-tune the last layer of the parametric model with the new training set. We also evaluated a mixture model, combining the classifications of the parametric model and the non-parametric model at decision level in the following manner:

$$p(y|q) = \lambda p_{\text{param}}(y|q) + (1 - \lambda)p_{\text{mem}}(y|q), \tag{5}$$

where the parameter $\lambda \in [0, 1]$ controls the contribution of each model (this model was proposed by Grave et al. (2016) in the context of language modelling). We created five random splits in new and old classes. Hyperparameters were tuned for all models using the first split and the validation set, and we report the average performance on the remaining splits evaluated on the test set.

Figure 4 shows the test set performance for all models, split by new and old classes.

While the mixture model provides a large improvement over the plain parametric model, MbPA significantly outperforms both of them both in speed and performance. This is particularly noticeable in the new classes, where MbPA acquires knowledge from very few examples. Table 1 shows a quantitative analysis of these observations. After around 30 epoches the parametric model matches the performance of MbPA. In the appendix we explore sensitivity of MbPA on this task to various hyperparameters (memory size, learning rate).

### 4.2.1 UNBALANCED DATASETS

We further explored the incremental introduction of new classes, specifically in the context of unbalanced datasets. Most real world data are unbalanced, whereas standard datasets (like ImageNet) are artificially balanced to play well with deep learning methods.

We replicated the setting from the ImageNet experiments in the previous section, where new classes were introduced to a pre-trained model. However, we only showed a tenth of the data for half the new classes and all data for the other half. We report performance on the full balanced validation set. Once again, we compared the parametric model with MbPA and a memory based mixture model.

Results are summarised in Figure 5 (left). After 20 epochs of training, MbPA outperformed both baselines, with a wider gap in performance than the previous experiment. Further, the mixture model, though equipped with memory, did significantly worse than MbPA, leading us to conclude that the inductive bias in the local adaptation process was well suited to deal with data scarcity.

| Subset | Model | Top 1 (at epochs) | | | AUC (at epochs) | | |
|---|---|---|---|---|---|---|---|
| | | 0.1 | 1 | 3 | 0.1 | 1 | 3 |
| Novel | MbPA | **46.2 %** | **64.5 %** | **65.7 %** | 27.4 % | **57.7 %** | **63.0 %** |
| | Non-Parametric | 40.0 % | 53.3 % | 52.9 % | **28.3 %** | 47.9 % | 51.8 % |
| | Mixture | 31.6 % | 56.0 % | 59.1 % | 18.6 % | 47.4 % | 54.7 % |
| | Parametric | 16.2 % | 53.6 % | 57.9 % | 5.7 % | 41.7 % | 51.9 % |
| Pre Trained | MbPA | 68.5 % | **70.9 %** | **70.9 %** | 71.4 % | 70.3 % | **70.3 %** |
| | Non-Parametric | 62.7 % | 69.4 % | 70.0 % | 45.9 % | 65.8 % | 68.7 % |
| | Mixture | **71.9 %** | 70.3 % | 70.2 % | 74.8 % | **70.6 %** | 70.1 % |
| | Parametric | 71.4 % | 68.1 % | 68.8 % | **76.0 %** | 68.6 % | 68.3 % |

Table 1: Quantitative evaluation of the learning dynamics for the Imagenet experiment. We compare a parametric model, non-parametric model (prediction based on memory only (9)), a mixture model and MbPA. We report the top 1 accuracy as well as the area under the curve (AUC) at different points in training.

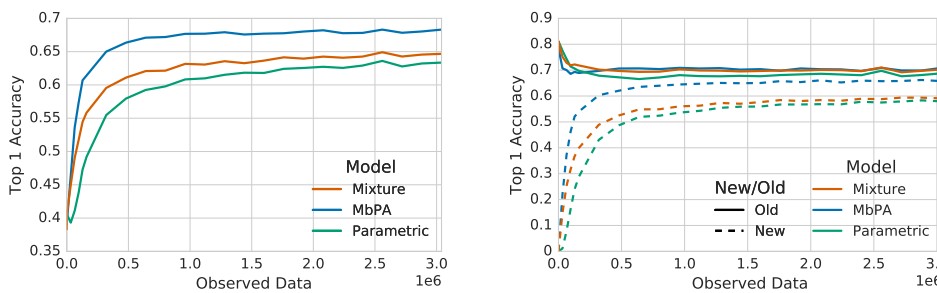

Figure 4: The figure compares the performance of MbPA (blue) against two baselines: the parametric model (green) and the mixture of experts (red). (Left) Aggregated performance (Right) disentangled performance evaluated on new (dashed) and old (solid) classes.

## 4.3 LANGUAGE MODELLING: DOMAIN SHIFTS

Finally we considered how MbPA can be used at test time to further improve the performance of language modelling. Given the general formulation of MbPA, this could be applied to any problem where there is a shift in distribution at test time — we focus on language modelling, where using recent information has proved promising, such as neural cache and dynamic evaluation (Grave et al., 2016; Krause et al., 2017).

We considered two datasets with established performance benchmarks, Penn Treebank (PTB; Marcus et al., 1993) and WikiText-2 (Merity et al., 2016). We pre-trained an LSTM and apply MbPA to the weights and biases of the output softmax layer. The memory stores the past LSTM outputs and associated class labels observed during evaluation. Full model details and hyper-parameters are detailed in Appendix B.

Penn Treebank is a small text corpus containing 887,521 train tokens, 70,390 validation tokens, and 78,669 test tokens; with a vocabulary size of 10,000. The LSTM obtained a test perplexity of 59.6 and this dropped by 4.3 points when interpolated with the neural cache. When we interpolated an LSTM with MbPA we were able to improve on the LSTM baseline by 5.3 points (an additional one from the cache model). We also attempted a dynamic evaluation scheme in a similar style to Krause et al. (2017), where we loaded the Adam optimisation parameters obtained during training and evaluated with training of the LSTM enabled, using a BPTT window of 5 steps. However we did not manage to obtain gains above 1 perplexity from baseline, and so we did not try it for WikiText-2.

WikiText-2 is a larger text corpus than PTB, derived from Wikipedia articles. It contains 2,088,628 train tokens, 217,646 validation tokens, and 245,569 test tokens, with a vocabulary of 33,278. Our LSTM baseline obtained a test perplexity of 65.9, and this is improved by 14.6 points when mixed with a neural cache. Combining the baseline LSTM with an LSTM fit with MbPA we see a drop of 9.9 points, however the combination of all three models (LSTM baseline + MbPA + cache) produced

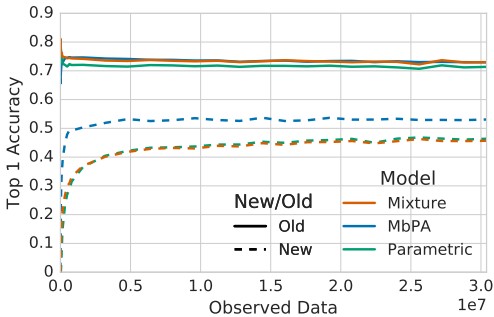 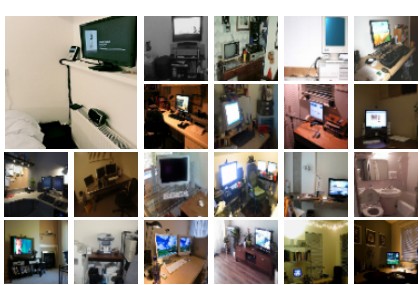

Figure 5: (Left) MbPA outperformed both parametric and memory-based mixture baselines, in the presence of unbalanced data on previously unseen classes (dashed lines). (Right) Example of MbPA. Query (shown larger in the top-right corner) of class "TV" and neighbourhood (all other images) for a specific case. Mixture and parametric models fail to classify the image while MbPA succeeds. 8 different classes in the closest 20 neighbours (e.g. "desktop computer", "monitor", "CRT screen"). Accuracy went from 25% to 75% after local adaptation.

the largest drop of 15.9 points. Comparing the perplexity word-by-word between LSTM + cache and LSTM + cache + MbPA, we see that MbPA improves predictions for rarer words (Figure 8).

| | PTB | | | WikiText-2 | | |
|---|---|---|---|---|---|---|
| | Valid | Test | $\Delta$Test | Valid | Test | $\Delta$Test |
| CharCNN (Zhang et al., 2015) | | 78.9 | | | | |
| Variational LSTM (Aharoni et al., 2017) | | 61.7 | | | | |
| LSTM + cache (Grave et al., 2016) | 74.6 | 72.1 | | 72.1 | 68.9 | |
| LSTM (Melis et al., 2017) | 60.9 | 58.3 | | 69.1 | 65.9 | |
| AWD-LSTM (Merity et al., 2017) | 60.0 | 57.3 | | 68.6 | 65.8 | |
| AWD-LSTM + cache (Merity et al., 2017) | 53.9 | 52.8 | - 4.5 | 53.8 | 52.0 | - 13.8 |
| AWD-LSTM *(reprod.)* (Krause et al., 2017) | 59.8 | 57.7 | | 68.9 | 66.1 | |
| AWD-LSTM + dyn eval (Krause et al., 2017) | 51.6 | 51.1 | - 6.6 | 46.4 | 44.3 | - 21.8 |
| LSTM (ours) | 61.8 | 59.6 | | 69.3 | 65.9 | |
| LSTM + cache (ours) | 55.7 | 55.3 | -4.3 | 53.2 | 51.3 | -14.6 |
| LSTM + MbPA | 54.8 | 54.3 | -5.3 | 58.4 | 56.0 | -9.9 |
| LSTM + MbPA + cache | 54.8 | 54.4 | -5.2 | 51.8 | 49.4 | -16.5 |

Table 2: Table with PTB and WikiText-2 perplexities. $\Delta$ Test denotes improvement of model on the test set relative to the corresponding baseline.

## 5   CONCLUSION

We have described Memory-based Parameter Adaptation (MbPA), a scheme for using an episodic memory structure to locally adapt the parameters of a neural network based upon the current input context. MbPA works well on a wide range of supervised learning tasks in several incremental, life-long learning settings: image classification, language modelling. Our experiments show that MbPA improves performance in continual learning experiments, comparable to or in many cases exceeding the performance of EWC. We also demonstrated that MbPA allows neural networks to rapidly adapt to previously unseen classes in large-scale image classification problems using the ImageNet dataset. Furthermore, MbPA can use the local, contextual updates from memory to counter and alleviate the effect of imbalanced classification data, where some new classes are over-represented at train time whilst others are underrepresented. Finally we demonstrated on two language modelling tasks that MbPA is able to adapts to shifts in word distribution common in language modelling tasks, achieving significant improvements in performance compared to LSTMs and building on methods like neural cache (Grave et al., 2016).

ACKNOWLEDGMENTS

We would like to thank Gabor Melis for providing the LSTM baselines on the language tasks. We would also like to thank Dharshan Kumaran, Jonathan Hunt, Olivier Tieleman, Koray Kavukcuoglu, Daan Wierstra, Sam Ritter, Jane Wang, Alistair Muldal, Nando de Frietas, Tim Harley, Jacob Menick and Steven Hansen for many helpful comments and invigorating discussions.

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

# A    MBPA HYPERPARAMETERS FOR INCREMENTAL LEARNING IMAGENET TASK

MbPA was robust and the inductive bias of the MbPA correction adapts the performance of the model on novel classes. This is shown in Figure 6 (right) where MbPA manages to achieve high performance almost at the same rate, regardless of the learning rate of the underlying parametric component.

In Figure 6 (left) we explore the influence in performance when changing the size of the episodic memory. We can see that the performance on the new classes is more sensitive to this parameter but it quickly saturates after about 400,000 entries. We repeat the above experiment by changing now the number of neighbours retrieved. The results are shown in Figure 7. We can observe that using more neighbours is better, but again, performance saturates quickly after 50 neighbours.

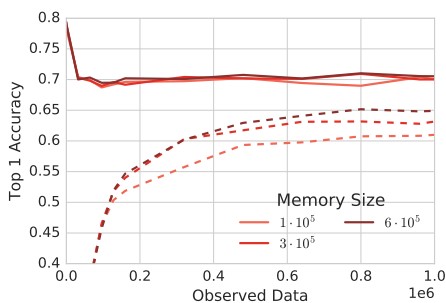
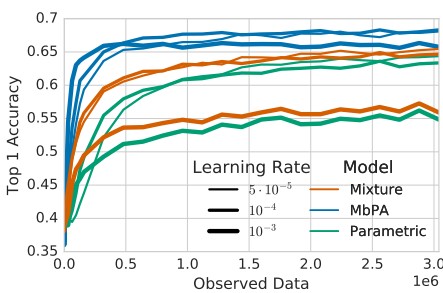

Figure 6: Left: Performance of MbPA when varying the dictionary size. Right: Performance of the parametric, mixture and MbPA models varying the learning rate of the parametric model. The colour code is the same as in Figure 4 and the thickness of the lines indicate the learning rate used.

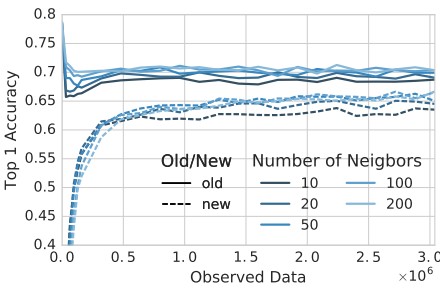

Figure 7: Performance of MbPA when varying the number of nearest neighours used for performing the local adaptation.

# B    MODEL DETAILS LANGUAGE MODELLING TASKS

For both datasets we used a single-layer LSTM baseline trained with Adam (Kingma & Ba, 2014) using the regularisation techniques described in Melis et al. (2017).

In this application of MbPA the test set is small (e.g. $< 80,000$ words for PTB), and so it was easy to overfit to the retrieved points. To remedy this, we tuned an L2 penalty $\beta \, ||\theta^x - \theta||_2$ term in our MbPA loss (7), where $\theta$ were the parameters derived from the training set and $\beta$ was a scalar hyper-parameter.

We swept over the following hyper-parameters:

- Memory size: $N \in \{500, 1000, 5000\}$

- Nearest neighbours: $K \in \{256, 512\}$
- Cache interpolation: $\lambda_{cache} \in \{0, 0.05, 0.1, 0.15\}$
- MbPA interpolation: $\lambda_{mbpa} \in \{0, 0.05, 0.1, 0.15\}$
- Number of MbPA optimisation steps: $T \in \{1, 5, 10\}$
- MbPA optimization learning rate: $\alpha \in \{0.01, 0.1, 0.15, 0.2, 0.5, 1\}$

Where memory size refers to both the MbPA memory size, and the size of the neural cache for comparison, and the $\lambda$ interpolation parameters refer to the mixing of model outputs, alike to Eq. 5. The optimal parameters were: $N = 5000$, $K = 256$, $\lambda_{cache} = 0.15$, $\lambda_{mbpa} = 0.1$, $T = 1$, $\alpha = 0.15$.

For Penn Treebank, we used a pre-trained LSTM baseline containing roughly $10M$ parameters with a hidden size of 1194 and a word embedding size of 268. For WikiText-2, we used a pre-trained LSTM baseline containing roughly 24M parameters with a hidden size of 1,853 and a word embedding size of 241.

## C  COMPARISON OF CACHE VS MBPA FOR WIKITEXT-2

The comparative benefit of MbPA is investigated, when combined with the LSTM + cache model. By computing the perplexity on a per-word basis and comparing whether the inclusion of MbPA improves (lowers) the perplexity, we can understand what types of words are better predicted. Anecdotal samples were not sufficient to understand the trend, however when the words were bucketed by their training frequency, we see a tend of improved performance for less frequent words.

This improved performance for rare words may be because the cache model has a prior to boost all recent words. Specifically, the cache probabilities are obtained from summing the attention for each instance of a word in memory, and so frequently occurring recent words that are not very contextually relevant will still be boosted. As MbPA does not do this, it appears to be more sensitive to infrequently occurring words.

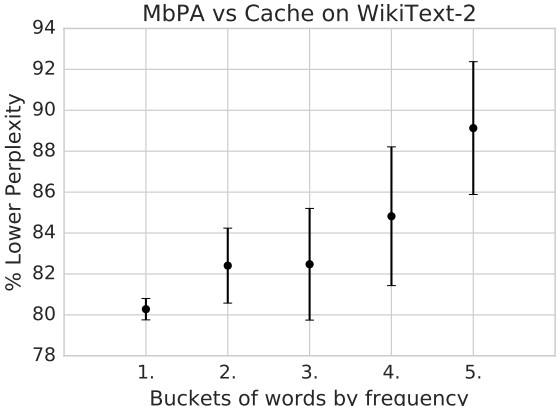

Figure 8: Percent improvement when MbPA is included with the LSTM baseline and neural cache, split by training word frequency into five equally sized buckets. The bucket 1 contains the most frequent words, and bucket 5 contains the least frequent words. The average improvement $\pm 1$ standard deviation are shown. MbPA provides a directional improvement for less frequent words.

## D  MAP INTERPRETATION OF MBPA AND DERIVATION OF CONTEXTUAL UPDATE

Let $x_c$ correspond to the input of the $h_c, v_c$ key-value pair in the context $C$ of a given input $x$. In other words, $h_c$ was computed by feeding $x_c$ to the embedding network. Then the posterior given

this pair and the parameter obtained after training $\theta$, can be written as:

$$p(\theta^x|\theta, x_c, v_c, x) = \frac{p(v_c|x_c, \theta^x, x)p(\theta^x|\theta)}{p(v_c|\theta, x_c, x)}. \tag{6}$$

If we maximise the posterior over the context $C$ with respect to $\theta^x$.

$$\arg\max_{\theta^x} \mathbb{E}_C \left\{ \log p(\theta^x|\theta, x_c, v_c, x) \right\} = \arg\max_{\theta^x} \ \log p(\theta^x|\theta) + \mathbb{E}_C \left\{ \log p(v_c|x_c, \theta^x, x) \right\}$$

$$= \arg\max_{\theta^x} \ \log p(\theta^x|\theta) + \sum_{k=1}^{K} w_k^{(x)} \log p(v_k^{(x)}|h_k^{(x)}, \theta^x, x). \tag{7}$$

Let $\log p(\theta^x|\theta) \propto -\frac{||\theta^x - \theta||_2^2}{2\alpha_M}$ (i.e. a Gaussian prior on $\theta^x$ centred at $\theta$) be thought as a regularisation term that prevents the local adaptation to move $\theta^x$ too far from $\theta$, preventing overfitting.

Another interpretation of (7) is that when the prior is taken to be a Gaussian, it is a form of elastic weight regularisation (similar to Kirkpatrick et al. (2017)) and the second term corresponds to the log likelihood of $\theta^x$ on the data in the context $C$. This can also be seen as posterior sharpening (Fortunato et al., 2017), where we can think of the second term as an approximation of $\log p(y_t|x_t, \theta)$. Thus a view of MbPA is it is a form of local elastic weight consolidation on a context dataset $C$.

Equation (7) does not have a closed form solution, and requires fitting a large number of parameters at inference time. This can be costly and susceptible to overfitting. We can avoid this problem by simply adapting the reference parameters $\theta$. Specifically, we perform a fixed number of gradient descent steps (or any of its popular variants) to minimise (7). One step of gradient descent to the loss in (7) with respect to $\theta^x$ yields

$$\Delta_M(x, \theta) = -\alpha_M \left. \nabla_\theta \sum_{k=1}^{K} w_k^{(x)} \log p(v_k^{(x)}|h_k^{(x)}, \theta^x, x) \right|_\theta - \beta(\theta - \theta^x), \tag{8}$$

where $\beta$ is a scalar hyper-parameter. These adapted parameters are used for output computation but discarded thereafter.

# E  ATTENTION AS A SPECIAL CASE OF MBPA

Let $C = \{(w_i, h_i, v_i)\}_{i=1}^{k}$ be the neighbourhood retrieved from memory given a query $q$. The likelihood prediction based on attention is given by

$$p_{\text{mem}}(y = j|q) = \frac{\sum_{i=1}^{k} w_i \delta(v_i = j)}{\sum_{i=1}^{k} w_i}, \tag{9}$$

where the Kronecker $\delta$ is one when the equality holds and zero otherwise. We now show how the attention-based prediction given in (9) can be seen as particular case of local adaptation.

For classification with $c$ classes, we parameterise $p_{\text{mem}}$ via its logits, $z \in \mathbb{R}^c$, with $p_{\text{mem}}(v|q) =$ softmax($z$). One good candidate $z$ is the one that is the most consistent with context $C$. Specifically, the logit vector that minimises the weighted average negative log-likelihood (NLL) of the memories in context $C$:

$$z_q = \arg\min_z \sum_{i=1}^{N} w_i \left( z_{v_i} - \log(\sum_{k=1}^{c} e^{z_k}) \right). \tag{10}$$

The attention weights scale the importance of each memory in the neighbour given its similarity to the query. This matches the loss (7) (ignoring the prior term). If we differentiate the above equation with respect to a $z_j$ and set to zero, we obtain exactly the same expression as in (9).

Effectively, in (10) we are fitting a constant function to the context retrieved from the episodic memory. This is a particular case of a local likelihood model (Loader, 2006). The update also is the same as applying a k-nn, see Figure 2.

Note that this interpretation is not limited to classification tasks, the exact same reasoning (and result) could be done for a regression task, simply by changing the loss function to be Mean Squared Error (MSE).

In this context, we can think of MbPA as a generalisation of the attention mechanism, in which the function used for the local fitting is given by the output network. Moreover, the parameters of the model are used as a prior for solving the local fitting problem and only change slightly to prevent overfitting.

