# OpenReview forum: "Memory-based Parameter Adaptation"
_ICLR.cc/2018/Conference — Accept (Poster)_

### Official Review · AnonReviewer3 · 2017-11-22
**The authors propose a memory-based mechanism to adapt model parameters with local context lookup.**

**Rating:** 6
**Confidence:** 4

**Review:**

Overall, the idea of this paper is simple but interesting. Via weighted mean NLL over retrieved neighbors, one can update parameters of output network for a given query input. The MAP interpretation provides a flexible Bayesian explanation about this MbPA.

The paper is written well, and the proposed method is evaluated on a number of relevant applications (e.g., continuing learning, incremental learning, unbalanced data, and domain shifts.)

Here are some comments:
1 MbPA is built upon memory. How large should it be? Is it efficient to retrieve neighbors for a given query?
2 For each test, how many steps of MbPA do we need in general? Furthermore, it is a bit unfair for me to retrain deep model, based on test inputs. It seems that, you are implicitly using test data to fit model.

---

> ### Author Response · Authors · 2017-12-15
> **Response to Questions from Reviewer3**
>
> Thank you for your review. Please find below our response and clarifications.
>
> 1) MbPA is built upon memory. How large should it be?
>
> * The optimal memory size is task dependent, but in general the larger the memory the better. However, performance saturates at a given point.
> * A nice property of the model (as shown in the continual and incremental learning setups) is that performance degrades gracefully as memory size decreases. For continual learning, even storing 1% of data seen on a task boosts performance significantly.
> * One important aspect to note is that the smaller the memory the more important it becomes to add regularization to prevent overfitting to the local context, as explained in Section 2.1. This is the case of the language modeling experiments.
> * For the ImageNet experiments we show how performance varies with memory size in Fig 6 (Appendix). We will include a similar evaluation for continual learning and language modeling tasks.
>
> 2)  Is it efficient to retrieve neighbors for a given query?
>
> * In this work it is the cost of an exact nearest neighbour search. Which is linear in memory size. We see that the cost of retrieving neighbours is negligible compared to the rest of the model (eg. the inner optimisation). For eg. on PTB language modeling with a cache size of 5000, the content based lookup is about ~20us, and each step of optimization is ~1ms on one GPU.
> * Fast approximate knn search can be used, but performance could degrade depending on the recall of the approximate search. This would be a nice direction for future work.
> * One of the advantages of not querying the memory at training time, is that we avoid this cost.
>
> 3) For each test, how many steps of MbPA do we need in general?
>
> * This is a hyper-parameter of the model. Across all tasks, we observed that a small number of iteration is sufficient, between 5 and 20. However, we see noticeable gains with even 1 step.
>
> 4) Furthermore, it is a bit unfair for me to retrain deep model, based on test inputs. It seems that, you are implicitly using test data to fit model.
>
> * Many algorithms have a clean split between train and test. They are unable to adapt to shifts in distribution. We are interested specifically in studying algorithms that are capable of adapting to domain shift. Or, to leverage the temporal correlation during an evaluation episode.
> * We only do this in the language model example, which deals with quickly adapting to a change in the data distribution at test time. The effect of online adaptation during test time has been long studied in this task, solutions dating back to Dynamic Evaluation (A. Graves’ thesis). Naturally, all these approaches use the test data in a causal way (as in online learning), meaning, only the examples that have been processed are available for training.
> * Note that we’re comparing with many models that also use the observed test samples to adapt their predictions. The data seen at each test example is thus consistent across all baselines.
>
> We will update the text to take into account all clarifications above.

---

### Official Review · AnonReviewer2 · 2017-11-26
**In general, I found this paper interesting but it needs improvement in writing and more clarity in the contribution. There are quite similarities (e.g. memory architecture and model components) with previous works which authors need to be more clear about their contributions.**

**Rating:** 6
**Confidence:** 5

**Review:**

This paper proposes a non-parametric episodic memory that can be used for the rapid acquisition of new knowledge while preserving the old ones. More specially, it locally adapts the parameters of a network using the episodic memory structure.

Strength:
+   The paper works on a relevant and interesting problem.
+   The experiment sections are very thorough and I like the fact that the authors selected different tasks to compare their models with.
+ The paper is well-written except for sections 2 and 3.
Weakness and Questions:
-    Even though the paper addresses the interesting and challenging problem of slow adaption when distribution shifts, their episodic memory is quite similar (if not same as) to the Pritzel et al., 2017.
- In addition, as the author mentioned in the text, their model is also similar to the Kirkpatrick et al., 2017,  Finn et al., 2017, Krause et al., 2017. That would be great if the author can list "explicitly" the contribution of the paper with comparing with those. Right now, the text mentioned some of the similarity but it spreads across different sections and parts.
- The proposed model does adaption during the test time, but other papers such as Li & Hoiem, 2016 handles the shift across domain in the train time. Can authors say sth about the motivation behind adaptation during test time vs. training time?
- There are some inconsistencies in the text about the parameters and formulations:
      -- what is second subscript in {v_i}_i? (page 2, 3rd paragraph)
      -- in Equation 4, what is the difference between x_c and x?
      -- What happened to $x$ in Eq 5?
      -- The "−" in Eq. 7 doesn't make sense.
- Section 2.2, after equation 7, the text is not that clear.
- Paper is well beyond the 8-page limit and should be fitted to be 8 pages.
- In order to make the experiments reproducible, the paper needs to contain full details (in the appendix) about the setup and hyperparameters of the experiments.

Others:
Do the authors plan to release the codes?


------------------------------------
------------------------------------
Update after rebuttal:
Thanks for the revised version and answering my concerns.
In the revised version, the writing has been improved and the contribution of the paper is more obvious.
Given the authors' responses and the changes, I have increased my review score.

A couple of comments and questions:
1. Can you explain how/why $x_c$ is replaced by $h_k$ in eq_7?
2. In the same equation (7), how $\log p(v_k| h_k,\theta_x, x)$ will be calculated? I have some intuition but not sure.  Can you please explain?
3. in equation(8), what happened to $x$ in log p(..)?
4. How figure 2 is plotted? based on a real experiment? if yes, what was the setting? if not, how?
5. It'd be very useful to the community if the authors decide to release their codes.

---

> ### Author Response · Authors · 2017-12-15
> **Response to Questions from Reviewer2 (1/2)**
>
> Thank you for your review. Please find below our response and clarifications. The comment has been split into two to ensure we are under the comment character limit.
>
> 1) Even though the paper addresses the interesting and challenging problem of slow adaptation when distribution shifts, their episodic memory is quite similar (if not same as) to the Pritzel et al., 2017.
>
> * Our memory module is indeed essentially the same as that of Pritzel et al, 2017, differing only on how the keys are obtained. The keys are embeddings computed from our parametric model (embedding + output networks) trained directly on the target task instead of relying on gradients through the memory. Note that several other works (cited in the manuscript) use very similar memory architectures. We do not claim the memory as one of our contributions, instead, the novelty lies in the use of the memory as a way of enhancing powerful parametric models. We will further clarify this in the text.
>
> 2) In addition, as the author mentioned in the text, their model is also similar to the Kirkpatrick et al., 2017,  Finn et al., 2017, Krause et al., 2017. That would be great if the author can list "explicitly" the contribution of the paper with comparing with those. Right now, the text mentioned some of the similarity but it spreads across different sections and parts.
>
> * We will include a detailed description of our contributions, and concentrate (and expand) the relation with previous work in Section 3.
> * The contributions of our work are: (i) proposing an architecture for enhancing powerful parametric models to include a fast adaptation mechanism to cope with changes in the task at hand (ii) we establish connections of our method with attention mechanisms frequently used for querying memories (iii) we present a bayesian interpretation of the method allowing a principled form regularization (iv) we evaluate the method in a range of different tasks: continual learning (pmnist), incremental learning (imagenet) and data distribution shifts (language), obtaining promising results.
> * The only similarity with Krause et al. is that we too use a memory buffer in the context of language modelling. Their method of using the memory is via a mixture of experts system to deal with recent words for language models. We do compare to this baseline for our LM experiments, however their method does not deal with the problem of distributional shifts and cannot be applied to continual or incremental learning set ups.
> * Finn et al. devises MAML - a way of doing meta-learning over a distribution of tasks. Both of our methods extend the classic fine-tuning technique used in domain adaptation type of ideas (e.g. fit a given neural network to a small set of new data). Their algorithm aims at learning an easily adaptable set of weights, such that given a small amount of training data for a given task following the training distribution, the fine-tuning procedure would effectively adapt the weights to this particular task. Their work does not use any memory or per-example adaptation and is not based on a continual (life-long) learning setting. In contrast, our work, aims at augmenting a powerful neural network with a fine-tuning procedure that is used at inference only. The idea is to enhance the performance of the parametric model while maintaining it's full training.
> * EWC, developed in Kirkpatrick et al. 2017 is powerful method of doing continual learning across tasks. The algorithm works by learning a new task with an additional loss forcing the model to stay close to the solution found on the previous task. This method makes no use of memory or local adaptation, requiring instead the storing of weights and fisher matrices for each task seen. We compare to this method for our continual learning tasks as a very competitive baseline. MbPA does not rely on storing past weights or fisher matrices. We show comparable performance with even 100 examples stored per task and show how these methods are orthogonal and can be combined. One similarity we do note is that adding a regularization term to the local loss of MbPA can be seen as a local version or approximation of the EWC loss term - i.e. forcing the model to stay close to the solution found at training time.

---

> ### Author Response · Authors · 2017-12-15
> **Response to Questions from Reviewer2 (2/2)**
>
>
> 3) The proposed model does adaptation during the test time, but other papers such as Li & Hoiem, 2016 handles the shift across domain in the train time. Can authors say sth about the motivation behind adaptation during test time vs. training time?
>
> * The work “Learning without forgetting” by Li & Hoiem is a simple and effective method for avoiding catastrophic forgetting. However, in our view, it doesn’t guarantee that the internal representations would be preserved and doesn’t show any evidence in this direction.
> * Our motivation is two-fold. First, we want our model to be able to consolidate the knowledge and be able to perform well without relying on the memory content. The memory then serves to boost performance by focusing the weights on memory relevant to the prediction at hand. Second, adapting the model during training is computationally very demanding (e.g. language modeling, imagenet).
> * Further, adaptation at test time for language modelling has strong established baselines such as Krause et al 2017. We thus wanted a comparable setting to the reported baselines.
>
> 4) Paper is well beyond the 8-page limit and should be fitted to be 8 pages.
>
> * ICLR has a soft page limit. We are aware that the text is long, but we didn’t want to leave out details on the experimental settings. We will take this comment into account and edit the text as needed after the clarifications mentioned here are added in, in an attempt to reduce the length of the paper by moving a few things into an appendix.
>
> 5) In order to make the experiments reproducible, the paper needs to contain full details (in the appendix) about the setup and hyperparameters of the experiments.
>
> * We currently include details on the hyper-parameter selection procedure, and provide the best performing options. We will further clarify if anything is missing and add details to the appendix.
>
> Further, thank you for pointing out typos and inconsistencies.
>
> * We will correct this in the paper and clarify subscripts and the text in section 2.2.
> * The negative sign in eq 7 is a typo.
> * x is the input being regressed or classified whereas x_c  ("c" is context; we will clarify this) is the input that was used to create embedding h_c stored in memory (with value v_c).
>
> We will update the text to take into account all clarifications above.

---

### Official Review · AnonReviewer1 · 2017-11-27
**Very interesting use of episodic memory, could be even stronger**

**Rating:** 8
**Confidence:** 4

**Review:**

This article introduces a new method to improve neural network performances on tasks ranging from continual learning (non-stationary target distribution, appearance of new classes, adaptation to new tasks, etc) to better handling of class imbalance, via a hybrid architecture between nearest neighbours and neural net.
After an introduction summarizing their goal, the authors introduce their Model-based parameter adaptation: this hybrid architecture enriches classical deep architectures with a non-parametric “episodic” memory, which is filled at training time with (possibly learned) encodings of training examples and then polled at inference time to refine the neural network parameters with a few steps of gradient in a direction determined by the closest neighbours in memory to the input being processed.  The authors justify this inference-time SGD update with three different interpretations: one linked in Maximum A Posteriori optimization, another to Elastic Weight Regularisation (the current state of the art in continual learning), and one generalising attention mechanisms (although to be honest that later was more elusive to this reviewer). The mandatory literature review on the abundant recent uses of memory in neural networks is then followed by experiments on continual learning tasks involving permuted MNIST tasks, ImageNET incremental inclusion of classes, ImageNet unbalanced, and two language modeling tasks.

This is an overall very interesting idea, which has the merit of being rather simple in its execution and can be combined with many other methods: it is fully compatible with any optimiser (e.g. ADAM) and can be tacked on top of EWC (which the authors do). The justification is clear, the examples reasonably thorough. It is a very solid paper, which this reviewer believes to be of real interest to the ICLR community.


The following important clarifications from the authors could make it even better:
*  Algorithm 1 in its current form seems to imply an infinite memory, which the experiments make clear is not the case. Therefore: how does the algorithm decide what entries to discard when the memory fills up?
* In most non-trivial settings, the parameter $gamma$ of the encoding is learned, and therefore older entries in the memory lose any ability to be compared to more recent encodings. How do the authors handle this obsolescence of the memory, other than the trivial scheme of relying on KNN to only match recent entries?
* Because gamma needs to be “recent”, this means “theta” is also recent: could the authors give a good intuition on how the two sets of parameters can evolve at different enough timescales to really make the episodic memory relevant? Is it anything else than relying on the fact that the lower levels of a neural net converge before the upper levels?
* Table 1:  could the authors explain why the pre-trained Parametric (and then Mixture) models have the best  AUC in the low-data regime, whereas MbPA was designed very much to be superior in such regimes?
* Paragraph below equation (5), page 3: why not including the regularisation term, whereas the authors just went to great pain to explain it? Rationale? Not including it is also akin to using an improper non-information prior on theta^x independent of theta, which is quite a strong choice to be made “by default”.
* The extra complexity of choosing the learning rate alpha_M and the number of  MpAB steps is worrying this reviewer somewhat. In practice, in Section 4.1the authors explain using grid search to tune the parameters. Is this reviewer correct in understanding that this search is done across all tasks, as opposed to only the first task? And if so, doesn’t this grid search introduce an information leak by bringing information from the whole pre-determined set of task, therefore undermining the very “continuous learning” aim? How do the algorithm performs if the grid search is done only on the first task?
* Figure 3:  the text could clarify that the accuracy is measured across all tasks seen so far. It would be interesting to add a figure (in the Appendix) showing the evolution of the accuracy *per task*, not just the aggregated accuracy.
* In the related works linking neural networks to encoded episodic memory, the authors might want to include the stream of research on HMAX of Anselmi et al 2014 (https://arxiv.org/pdf/1311.4158.pdf) , Leibo et al 2015 (https://arxiv.org/abs/1512.08457), and Blundell et al 2016 (https://arxiv.org/pdf/1606.04460.pdf ).

Minor typos:
* Figure 4: the title of the key says “New/Old” but then the lines read, in order, “Old” then “New” -- it would be nicer to have them in the same order.
* Section 5: missing period between "ephemeral gradient modifications" and "Further".
* Section 4.2, parenthesis should be "perform well across all 1000 classes", not "all 100 classes".

With the above clarifications, this article could become a very remarked contribution.

---

> ### Author Response · Authors · 2017-12-15
> **Response to Questions from Reviewer1 (1/2)**
>
> Thank you for your review. Please find below our response and clarifications. The responses have been split to ensure we are under the ICLR comment character limit.
>
> 1) Algorithm 1 in its current form seems to imply an infinite memory, which the experiments make clear is not the case. Therefore: how does the algorithm decide what entries to discard when the memory fills up?
>
> * In the current implementation we simply treat the memory as a circular buffer, in which we overwrite the oldest data as the memory gets full. We will clarify this on the text.
> * Deciding what to store (or overwrite) is indeed a very interesting question that we did not explore and will address in future work. We evaluated a few heuristics (e.g. storing only examples with high training loss) that did not perform better than the circular buffer described above.
>
> 2) In most non-trivial settings, the parameter $gamma$ of the encoding is learned, and therefore older entries in the memory lose any ability to be compared to more recent encodings. How do the authors handle this obsolescence of the memory, other than the trivial scheme of relying on KNN to only match recent entries?
>
> * Having a stable (or slow changing) network is important for being able to have long term recall. This could be justified (as the reviewer mentions) by the fact that lower level parameters converge faster than those in the higher part of the network. Hence, it is inevitable some memory obsolescence in the beginning of training. This is also the case on humans as infant amnesia could be explained as memories stored with an old (not consolidated) network that cannot be recovered later in life. We will include a short comment further clarifying this important point.
> * An alternative approach would be to rely on replay of raw data (e.g. store the input images from pixels). A downside is that, unlike internal activations (embeddings), replaying raw data requires a large amount of storage. However many artificial systems do it (e.g. DQN for RL). If we store raw data, we could still base our look-ups on a distance in the embedding space in order to obtain a semantic (more relevant) metric. We would replay the memories to prevent catastrophic forgetting and periodically recompute the embeddings to keep them up to date. We did not implement this variant.
>
> 3) Table 1:  could the authors explain why the pre-trained Parametric (and then Mixture) models have the best  AUC in the low-data regime, whereas MbPA was designed very much to be superior in such regimes?
>
> * Note that this happens only for the classes that were used during pre-training. The result makes sense: the initial parametric model performs very well on the classes that was pre-trained on. The memory is initially empty, so adapting the predictions of parametric model (via MbPA or the mixture model) using few examples slightly degrades its performance in the beginning. This quickly changes as more examples are collected.
> * On the other hand, for the new classes, relying on the memories massively improves performance even when few examples have been stored.
>
> 4) Paragraph below equation (5), page 3: why not including the regularisation term, whereas the authors just went to great pain to explain it? Rationale? Not including it is also akin to using an improper non-information prior on theta^x independent of theta, which is quite a strong choice to be made “by default”.
>
> * We wrote it in this way for ease of explanation and developed later in Section 2.1, as we only talk about the bayesian interpretation then. We will change the text accordingly.

---

> ### Author Response · Authors · 2017-12-15
> **Response to Questions from Reviewer1 (2/2)**
>
>
>
> 5) The extra complexity of choosing the learning rate alpha_M and the number of  MbPA steps is worrying this reviewer somewhat. In practice, in Section 4.1 the authors explain using grid search to tune the parameters. Is this reviewer correct in understanding that this search is done across all tasks, as opposed to only the first task? And if so, doesn’t this grid search introduce an information leak by bringing information from the whole pre-determined set of task, therefore undermining the very “continuous learning” aim? How do the algorithm performs if the grid search is done only on the first task?
>
> * We agree with the reviewer: setting the hyper-parameters would leak information from the future tasks. We do not do this in our experiments.
> * The hyper-parameters were obtained using different variants of permuted MNIST, following the standard practice for continual learning.
> * It is worth noting that we empirically found that MbPA is not very sensitive to the choice other parameters such as inner learning rate or number of steps (especially when combined with the regularization term or EWC). The tuning was required more for the EWC baseline where there is a tradeoff between learning new tasks and remembering old ones based on the weighting of the loss. For MbPA for CL we found any number of steps between 5-10 worked well with high learning rates between 0.1 and 1.0.
> * For MbPA, we reported several hyper-parameters (i.e memory size) to give a feel of the sensitivity of the algorithm.
>
> 6) Figure 3:  the text could clarify that the accuracy is measured across all tasks seen so far. It would be interesting to add a figure (in the Appendix) showing the evolution of the accuracy *per task*, not just the aggregated accuracy.
>
> * We will include this figure and clarify this in the text.
> * For the EWC baseline we find that (as mentioned above) the per-task curves are very different based on which tasks you care about more (e.g trivially setting the EWC penalty to a high value would give near perfect accuracy on the first task and no learning on the others). The only way to tune is to in fact, look at final average accuracy on (another, validation set) of permuted pixels and then apply that to the final test version. For MbPA, we found it shows gradual forgetting across tasks based on how many examples are stored per task.
>
> 7) In the related works linking neural networks to encoded episodic memory, the authors might want to include the stream of research on HMAX of Anselmi et al 2014 (https://arxiv.org/pdf/1311.4158.pdf) , Leibo et al 2015 (https://arxiv.org/abs/1512.08457), and Blundell et al 2016 (https://arxiv.org/pdf/1606.04460.pdf ).
>
> * Thank you for the links to relevant work - we will include all these references.
>
> We will update the text to take into account all clarifications above and typos mentioned.

---

> ### Comment · AnonReviewer1 · 2018-01-12
> **Response to Response**
>
> Dear Authors and AC
>
> Thank you for your detailed answers -- having to split in two comments due to length shows how seriously you take it :)
> Between them and the fact that my mind kept wandering back to the ideas in this paper during the holidays, I am happy to maintain my score of 8 - Top 50% papers.

---

### Public Comment · (anonymous) · 2017-11-22
**Few Questions**

1) You compare against the cache model of Grave et al, however their results depend on the size of the cache.
The comparisons doesn't seem fair, as the neural cache has bounded memory, but I didn't catch the amount of memory deployed in your model. Furthermore as your results suggest that for a given LSTM that neural cache seemed to be better. Can you provide some details about the size of cache used versus the amount of memory your model used.

2) Since the method seems very related to MAML (Finn, Abbeel, and Levine 2017), a comparison against it would be good to see, where both methods are applicable.

3) There has been other recent work on online modeling and adaptation using history/memory.  A discussion about relevance/similarity/differences between your model wrt these models would be great.

Unbounded cache model for online language modeling with open vocabulary
https://arxiv.org/pdf/1711.02604.pdf by Grave, Cisse and Joulin

Improving One-Shot Learning through Fusing Side Information
https://arxiv.org/pdf/1710.08347.pdf by Tsai and Salakhutdinov
(This one uses attention on history, and doesn't seem fundamentally different from memory)

Meta-Learning via Feature-Label Memory Network
https://arxiv.org/pdf/1710.07110.pdf by Mureja, Park and Yoo

Label Organized Memory Augmented Neural Network
https://arxiv.org/pdf/1707.01461.pdf by Shankar and Sarawagi

Online Adaptation of Convolutional Neural Networks for Video Object Segmentation
https://arxiv.org/pdf/1706.09364.pdf by Voigtlaender and Leibe

---

> ### Author Response · Authors · 2017-11-23
> **Reply: Few Questions**
>
> Thanks for your comment. Please see below our response.
>
> (1) A valid point is made about memory constraints, which can be clarified in the text. When comparing MbPA against the Neural Cache we do use a bounded memory in both cases. We swept over memory sizes from 500 - 20,000. E.g. for WikiText2 the optimal memory size for the neural cache was 6,000, whereas for MbPA it was 5,000. We will update the paper with further details of hyper-parameter sweeps and optimal parameters. For the language modeling experiments it is apparent the unbounded cache, which was posted on arXiv after the ICLR submission date, performs strictly worse than the bounded cache and so we prefer to compare our method versus the best performing variant. As for the point about performance for a given LSTM: on PTB, MbPA produced the best results in our experiments. On WikiText2, we find that MbPA does not focus purely on most recent terms and thus the combination of all three models (LSTM baseline, MbPA and neural cache) produced the largest drop of 15 perplexity.
>
> (2) MbPA and MAML are indeed similar in that they aim to fine-tune or adapt a network with relevant data. The general idea of adapting a network to a relevant context is also not unique to these methods but an old idea: speaker adaptation, dynamic evaluation, Dyna2, etc.
> The contribution of MAML is to train a base model through the task-specific optimization, to obtain a set of ‘easily tunable’ parameters that can be adapted to the task at hand, while MbPA adapts its parameters in an online fashion using its episodic memory.
>
> In the continual learning setup (eg. permuted MNIST) - one does not have the ability to re-train on a previous task and thus there is then no way of applying MAML in this setting without access to this task oracle (which would give MAML privileged information).
>
> The same is true for language modeling where there are in fact no clear “tasks”, but a changing data distribution at test time which MbPA’s context based lookup can cope with. Instead for LM we compare MbPA to dynamic evaluation, which is a more relevant alternative method of local fitting. Conversely, applying MbPA to the MAML tasks would not work well on the first visit, as its memory would have no experience from this un-seen task, but would be able to cope with the task at test time.
>
> (3) This is a great list of references that we will be sure to describe it in an expanded literature review. Most however seem to have been published around or after the ICLR deadline and thus have not been addressed. We discuss the unbounded cache in comment (1) above.

---

> > ### Public Comment · (anonymous) · 2017-12-27
> > **Re: Reply: Few Questions**
> >
> > If MbPA has a circular buffer of same size as Neural cache, I don't clearly understand how the 'focus purely on most recent terms' works. Focus on recent terms make sense as in their results the benefit of increasing memory fell very quickly. But then a small memory in MbPA should be able to match that. So perhaps some tweaking of parameters might improve results?
> >
> > It will be interesting to apply this technique to some applications I am looking at, I hope the code becomes available soon

---

> > > ### Author Response · Authors · 2018-01-05
> > > **MbPA and Cache Model**
> > >
> > > We have rephrased this as the wording was ambiguous. What we found is that the cache model and MbPA benefit slightly different types of words. Namely, MbPA is better at predicting less frequent words. Although they both operate on the same recent past, we think this is because the cache model sums over its attention on a per-word basis, which means that frequently occuring not-so-relevant words in the cache are boosted. Whereas MbPA optimizes over a neighbourhood of K nearest neighbours. We have added this set of results to the appendix. In the context of language modelling, boosting recent words is quite a strong structural prior in itself [1] and so we don’t expect MbPA to necessarily improve performance over a cache but the combination is certainly more powerful.
> > >
> > > [1] Speech recognition and the frequency of recently used words: a modified Markov model for natural language. Kuhn, Roland. 1988

---

> > > > ### Public Comment · (anonymous) · 2018-01-05
> > > > **Comparing MbPA and cache**
> > > >
> > > > I think I understand the difference, however I still don't see exactly why one cannot match the other.
> > > > If a word occurs twice in the memory buffer, then gradient computed based on memory will get more strongly influenced by those terms. The relative weight is bounded by 1 in MbPA compared to  cache, however the on the other hand, its directly changing the network weights which has non-linear effects on prediction. It is likely a question of empirics.

---

### Comment · Area_Chair · 2017-12-13
**A few missing references**

there have been a few work from neural machine translation that are highly relevant to this work, such as

https://arxiv.org/abs/1609.06490: per-example adaptation based on nearest neighbours without weighting
https://arxiv.org/abs/1705.07267: memory-based nearest neighbour use without online adaptation of the parameters

it would be good to see both (1) the discussion on how the proposed approach is more advantageous over these prevoius work (one of them over 1 year old), and (2) how the proposed approach compares against them.

there's a more recent related work (so perhaps no need to compare against) in

https://arxiv.org/abs/1709.08878

it would be nice to discuss how it's related, especially since the paper above conducted experiments on language modelling (similarly to this submission.)

Note that this is not a meta review but just a comment.

---

> ### Author Response · Authors · 2017-12-15
> **Response to Comment**
>
> Thank you for comment and links to relevant work. Please find our response below (in order of the papers linked).
>
> (1) "One Sentence One Model for Neural Machine Translation":
> * We were not aware of this work. Thank you for the reference. The method is indeed very related in spirit to our approach, however it has several important differences.
>
> * The objective of their work is, given a test sentence, to bias the model by fine-tuning it on a relevant context available in the training set. The crucial difference is that our method concentrates on enhancing powerful parametric models to include a fast adaptation mechanism to cope with changes in the task at hand (i.e. when task itself changes). For this reason, our work tackles a larger range of problems than (1). Further, on incremental and continual learning we assume we do not have access to the original training data again and thus (1) would not be applicable. The use of memory in MbPA alleviates the need for access to the (potentially large) training data. Below, we address the comparison in the language domain.
>
> * Locally fitting a context retrieved from a training set is only useful when the train and test distributions are the same. As the authors of (1) state, their method becomes very good when highly similar sentences are available in the training set. In the case of language modeling, given a partial sentence, the distribution over next words is naturally multi-modal. The important point of our approach is to quickly leverage on the information available in the observed part of test set in an online fashion, to capture distributional shifts (e.g. specifics of the writing style, frequent use of some specific set of words) to disambiguate similar contexts present in the training data.
>
> * Another setting in which (1) could play an important role is when the capacity of the model is not large enough to properly fit the training set. For instance, replicating the approach in (1) in the ImageNet task: if we fully train a ResNet on the whole training set and then apply the local fitting at test time (as in (1)), very little gain would be observed. As it is rare to find very similar images in the training set and the plain parametric model archives very high performance on the training set (about 95% top 1).
>
> * Other differences are: the weighting of the neighbors and the regularization term (obtained from our bayesian interpretation) to prevent overfitting the local neighbourhood.
>
> (2) "Search Engine Guided Non-Parametric Neural Machine Translation"
>
> * Thank you for raising this paper, we will certainly discuss similarities and differences to it in our related work. The core contribution of our paper, we feel, is how to incorporate information from memory into the final model predictions. In Gu et al. there are many interesting contributions, namely that of combining a non-differentiable search engine, differentiable attention, and incorporation of retrieved words into final predictions. The authors find that shallow mixing, i.e. interpolating probability distributions from memory and model, works best. We show in this paper that memory-based parameter adaptation is another competitive strategy to shallow mixing, and often working better (PTB for language modeling, ImageNet for image classification). As such, MbPA could be slotted into the search-engine guided translation model --- but we think this is best left for subsequent research projects.
>
> We will update the paper to take into account these references and expand the literature review.

---

### Author Response · Authors · 2018-01-05
**Revision to Submission**

Thank you all for the very helpful comments on our submission and links to related works.

To this end, we have uploaded a revised version incorporating changes mentioned in the comments, reviews and replies below.  This includes more detailed comparisons with other works, concretely outlining our contributions, details of hyper parameter selection and an expanded reference section. Various typos have been fixed and clarifications added.

We look forward to further comments and discussions.

---

### Public Comment · (anonymous) · 2018-02-01
**A few clarifications.**

Although the paper is written in a in very comprehensible manner I would like to ask the authors for a few clarifications:
1. What is theta_j in Eq (4)?
2. In Eq (4) what is p(v_k| h_k, theta_j)? How should one calculate it?
3. If the memory is treated as a circular buffer the oldest entries are overwritten as the memory gets full. Does is it mean that the model forgets early distributions entirely?

---

### Decision · Program_Chairs · 2018-01-29
**ICLR 2018 Conference Acceptance Decision**

**Decision:**

Accept (Poster)

**Comment:**

the proposed approach nicely incorporates various ideas from recent work into a single meta-learning (or domain adaptation or incremental learning or ...) framework. although better empirical comparison to existing (however recent they are) approaches would have made it stronger, the reviewers all found this submission to be worth publication, with which i agree.